# Prognostic and Therapeutic Potential of the OIP5 Network in Papillary Renal Cell Carcinoma

**DOI:** 10.3390/cancers13174483

**Published:** 2021-09-06

**Authors:** Mathilda Jing Chow, Yan Gu, Lizhi He, Xiaozeng Lin, Ying Dong, Wenjuan Mei, Anil Kapoor, Damu Tang

**Affiliations:** 1Department of Surgery, McMaster University, Hamilton, ON L8S 4K1, Canada; zhouj32@mcmaster.ca (M.J.C.); guy3@mcmaster.ca (Y.G.); linx36@mcmaster.ca (X.L.); dongy87@mcmaster.ca (Y.D.); wenjuanmei1986@gmail.com (W.M.); 2Urological Cancer Center for Research and Innovation (UCCRI), St Joseph’s Hospital, Hamilton, ON L8N 4A6, Canada; 3The Research Institute of St Joe’s Hamilton, St Joseph’s Hospital, Hamilton, ON L8N 4A6, Canada; 4Cutaneous Biology Research Center, Harvard Medical School and Massachusetts General Hospital, Boston, MA 02115, USA; LHE3@mgh.harvard.edu

**Keywords:** OIP5, papillary renal cell carcinoma, PLK1, tumorigenesis, therapy, biomarkers

## Abstract

**Simple Summary:**

Papillary renal cell carcinoma (pRCC) is an aggressive kidney cancer. Currently, there are no effective prognostic biomarkers and lack of efficacious therapies in treating pRCC. We report a novel and critical pRCC oncogenic factor OIP5. Its expression is increased in pRCC and the upregulation is associated with adverse features. High levels of OIP5 effectively predict pRCC recurrence and fatality. OIP5 promotes pRCC cell proliferation and tumor formation through complex processes. A 66-gene multigene panel (Overlap66) was constructed. Overlap66 is novel and robustly predicts pRCC recurrence and fatality. High risk pRCCs stratified by Overlap66 are associated with immune suppression. Furthermore, PLK1 is a component gene of Overlap66; PLK1 inhibitor significantly reduced OIP5-promoted pRCC cell proliferation in vitro and tumor growth in vivo. Collectively, Overlap66 can effectively stratifies high-risk pRCCs and these tumors can be treated with PLK1 inhibitors. Our findings can be explored for personalized therapy in pRCC patients.

**Abstract:**

Papillary renal cell carcinoma (pRCC) is an aggressive but minor type of RCC. The current understanding and management of pRCC remain poor. We report here OIP5 being a novel oncogenic factor and possessing robust prognostic values and therapeutic potential. OIP5 upregulation is observed in pRCC. The upregulation is associated with pRCC adverse features (T1P < T2P < CIMP, Stage1 + 2 < Stage 3 < Stage 4, and N0 < N1) and effectively stratifies the fatality risk. OIP5 promotes ACHN pRCC cell proliferation and xenograft formation; the latter is correlated with network alterations related to immune regulation, metabolism, and hypoxia. A set of differentially expressed genes (DEFs) was derived from ACHN OIP5 xenografts and primary pRCCs (*n* = 282) contingent to OIP5 upregulation; both DEG sets share 66 overlap genes. Overlap66 effectively predicts overall survival (*p* < 2 × 10^−16^) and relapse (*p* < 2 × 10^−16^) possibilities. High-risk tumors stratified by Overlap66 risk score possess an immune suppressive environment, evident by elevations in Treg cells and PD1 in CD8 T cells. Upregulation of PLK1 occurs in both xenografts and primary pRCC tumors with OIP5 elevations. PLK1 displays a synthetic lethality relationship with OIP5. PLK1 inhibitor BI2356 inhibits the growth of xenografts formed by ACHN OIP5 cells. Collectively, the OIP5 network can be explored for personalized therapies in management of pRCC patients.

## 1. Introduction

Renal cell carcinoma (RCC) accounts for approximately 85% of kidney cancer cases. RCC can be classified as clear cell RCC (ccRCC, 75%) and non-ccRCC (nccRCC, 25%) [1]. In the latter group, papillary RCC (pRCC) constitutes 50–64% of total incidence [2,3]. Morphologically, pRCC consists of two subtypes: type 1 pRCC (T1P) and type 2 pRCC (T2P) [4]. T1P and T2P are often associated with low nuclear grade (Fuhrman 1–2) and high nuclear grade (Fuhrman 3–4) tumors respectively [4,5], providing a clinical basis for T2P tumors having poor prognosis [6,7,8,9]. Genetically, while T1P tumors typically have alterations in the *MET* gene leading to abnormal MET activation [10], T2P tumors are heterogenous and contains: (1) mutations in the *FH* (fumarate hydratase) [11], *CDKN2A*, *SETD2*, *BAP1*, and *PBRM1* genes [12], (2) CpG island methylator phenotype (CIMP), and (3) activation of the NFR2-ARE (antioxidant response element) pathway [12]. Among the T2P tumors, CIMP subtype show a particularly low possibility of overall survival [12].

While these morphological and molecular subtyping offers a primary prognostic assessment of pRCC, significant improvement is needed to enhance patient counselling and management. Effective prediction of the risk of pRCC relapse is essential in offering personalized treatments; this risk assessment is particularly important in the light that surgery remains the primary treatment for localized pRCC with a relapse rate of nearly 40% [13]. Furthermore, therapeutic options for recurrent and metastatic pRCCs are limited and non-effective, which was partly a result of treatments being extrapolated from ccRCC studies. For instance, sunitinib is a standard of care for patients with metastatic pRCC [14], despite the therapeutic benefits being low and not as effective as for ccRCC [15]. The lack of effective prognostic biomarkers and therapeutic options highlight the unmet need for a more thorough investigation of the critical factors regulating pRCC progression.

Opa interacting protein 5 (OIP5) was discovered as an Opa (*Neisseria gonorrhoeae* opacity- associated) interacting protein [16]. The protein is highly enriched in human testis (https://www.proteinatlas.org/ENSG00000104147-OIP5/tissue, accessed on 1 August 2021) [17]; its upregulation is associated with adverse clinical features in multiple cancer types, including leukemia [18], ccRCC [19], glioma [20], and the cancers of the liver [21,22], lung [23], breast [24], gastric [25,26], and bladder [27,28,29,30]. Functionally, knockdown of OIP5 was reported to attenuate the proliferation of bladder cancer cells [29], as well as colorectal and gastric cells in vitro [25]. Building on these limited studies (*n* = 17 articles in PubMed on 5 April 2021) reporting a relevance of OIP5 in oncogenic events, much more remains unanswered for OIP5-facilitated oncogenesis, particular in the context of pRCC, as OIP5 has yet to be reported in studies related to pRCC.

We provide the first comprehensive analysis of OIP5’s oncogenic contributions in pRCC. OIP5 expression is significantly upregulated in pRCC; high levels of OIP5 correlate with adverse clinical characteristics of the disease, including stage, histological subtype (T2P), molecular subtype (CIMP), and lymph node metastasis. OIP5 expression robustly stratifies the risk of pRCC progression (progression-free survival) and fatality (overall survival and disease-specific survival). Functionally, OIP5 promotes pRCC cell proliferation in vitro and xenograft growth in vivo. Mechanistically, OIP5 facilitates pRCC progression along with network alterations; these changes show robust prognostic efficacies for rapid pRCC progression and fatality risk. Those of high-risk tumors display alterations in immune cell subsets including increases in the regulatory T (Treg) cell population. Treg cells are a major contributor to tumor-associated immune suppression [31]. Additionally, we identified polo-like kinase 1 (PLK1) as an OIP5-related gene in pRCC; the inhibition of PLK1 reduced OIP5-derived promotion of pRCC xenograft growth in vivo. Collectively, we report here (1) novel multigene sets derived from the OIP5 network that effectively predict the shortening of progression-free survival (PFS), overall survival (OS), and disease-specific survival (DSS) of pRCC, (2) an immune suppressive environment in pRCC tumors with OIP5 upregulation, and (3) inhibition of PLK1 as a potentially effective therapy in pRCC harboring OIP5 upregulation.

## 2. Materials and Methods

### 2.1. Cell Lines, Plasmid, and Retrovirus Infection

ACHN pRCC cell line and 786-O ccRCC cell line were purchased from ATCC and cultured in MEM and RPMI1640 respectively (Gibco, Carlsbad, CA, USA), both supplemented with 1% Penicillin-Streptomycin (Gibco, Carlsbad, CA, USA) and 10% fetal bovine serum (Life Technologies, Burlington, ON, USA). Cell lines were routinely checked for Mycoplasma contamination using a PCR kit (Abm, Cat#: G238). OIP5 cDNA plasmid was obtained from Origene (Cat: RG202255, Rockville, MD, USA) and subcloned into pBABE-puro retroviral plasmid (From Dr. Tak Mak at University of Toronto). Packing of retrovirus and the subsequent transfection were performed following our published conditions [32].

### 2.2. Invasion and Soft Agar Assay

Insert chambers with a control or matrigel membrane (8 µM pore size) for 24-well plates (Life sciences Corning^®^ BioCoat™, Glendale, AZ, USA) was used for invasion assay following manufacturer’s instructions. Cells (10^4^) were seeded into the top chamber; serum-free medium and 10% serum medium was added to the top and bottom chamber, respectively. Cells passing through the membrane were stained with crystal violet (0.5%). Soft agar assay was performed following our published conditions [32].

### 2.3. Colony Formation Assay and Proliferation Assay

Growth curves were generated by seeding 10^5^ cells/per well into 6-well tissue culture plates. Cell numbers were counted every 2 days. Colony formation assay was conducted by seeding cells in six-well plates with 100, 500, 1000 cells for ACHN, and 100, 300, 500 for 786-O. Colonies were fixed by fixation buffer (2% formaldehyde) and stained by crystal violet (0.5%) after cultured for 2 weeks. Colony numbers were counted and analyzed.

### 2.4. Western Blot

Cell lysates were prepared, and western blot was carried out as we have previously published [32]. Antibodies used included Anti-Flag M2 (1:1500, Sigma-Aldrich, Oakville, ON, Canada) and Anti-OIP5 (1:500, Sigma-Aldrich, Oakville, ON, Canada).

### 2.5. Immunohistochemistry (IHC)

Kidney cancer TMA (KD29602) was purchased from US Biomax (Dervood, MD, USA). Slide was baked at 60 °C for 1 h, then de-paraffinized in 100% xylene and 70% EtOH series. Antigen retrieval buffer was prepared with sodium citrate buffer (PH = 6) in the steamer for 20 min. OIP5 (1:50, Sigma-Aldrich, Oakville, ON, Canada) antibodies were incubated at 4 °C overnight. Secondary anti-rabbit antibodies (Vector Laboratories, 1:200), VECTASTAIN ABC and DAB solution (Vector Laboratories) were subsequently added to the slides and incubated following our IHC protocol. Washes were performed by 1× PBS and distilled water. Slides were counterstained by haematoxylin (Sigma Aldrich, Oakville, ON, Canada) and image analysis was conducted with ImageScope software (Leica Microsystems Inc., Richmond Hill, ON, Canada). Staining intensity scores were calculated into HScore by the formula [HScore = (%Positive) × (Intensity) + 1]. Statistical analysis was performed by student *t*-test, and *p* < 0.05 was considered statistically significant.

Xenograft tumors were paraffin embedded and cut serially by microtome. OIP5 (1:50, Sigma-Aldrich), Anti-Phospho-Histone H3 (Ser 10) (1:200, Upstate Biotechnology Inc., Lake Placid, NY, USA), CDK2 (1:200, Santa Cruz, Dallas, TX, USA), and PLK1 (1:300, Novus Biologicals, Toronto, ON, Canada) antibodies were used in the analyses for the xenograft tumors.

### 2.6. Xenograft Tumor Formation and Treatment with PLK1 Inhibitor

ACHN OIP5 and ACHN EV were suspended in 0.1 mL MEM/Matrigel (BD) mixture with 1:1 volume and implanted subcutaneously into the left flank of 8-week-old non-obese diabetic/severe combined immunodeficiency (NOD/SCID) male mice (The Jackson Laboratory). The mice were monitored post-injection of cancer cells through observation and palpation. The size of the tumors was measured every two days by caliper. Tumor volume was calculated based on the formula V = L × W2 × 0.52. BI2536 PLK1 inhibitor (Selleckchem, Burlington, ON, Canada) was dissolved in 0.1 N HCl and diluted by 0.9% NaCl. Diluted BI2536 or 0.9% NaCl (negative control) was injected to mice intravenously via tail vein with a dosage of 50 mg/kg. The mice were euthanized when the tumor volume reached 1000 mm^3^. The xenograft tumor, together with all the major organs, were photographed and collected. All tumors were cut in half, with one half fixed with 10% formalin (VWR, Mississauga, ON, Canada) and the other half stored in −80 °C. The formalin-fixed tissue was processed by department of Histology (St. Joseph’s Health care, Hamilton, ON, Canada) and embedded in paraffin. All the animal works were performed according to the protocols approved by McMaster University Animal Research Ethics Board (16-06-23).

### 2.7. RNA Sequencing Analysis

RNA sequencing analysis was carried out following our established conditions [33]. RNA was extracted from ACHN EV (*n* = 3) and ACHN OIP5 (*n* = 3) xenografts using a miRNeasy Mini Kit (Qiagen, No. 217004) according to the manufacturer’s instructions. RNA-seq libraries were generated with TruSeq Ribo Profile Mammalian Kit (Illumina, RPHMR12126) according to manufacturer’s instruction. These libraries were sequenced in a paired end setting by Harvard Bauer Core Facility using Nextseq 500/550. RNA-seq reads were processed and analyzed using Galaxy (https://usegalaxy.org/, accessed on 31 May 2020). Specifically, low quality reads and adaptor sequences (AGATCGGAAGAGCACACGTCTGAACTCCAGTCA: forward strand and AGATCGGAAGAGCGTCGTGTAGGGAAAGAGTGT: reverse strand) were first removed. Alignment and read counts were performed using HISAT2 and Featurecounts respectively. Differential gene expression was determined using DESeq2. KEGG analysis and GSEA (Gene Set Enrichment Analysis) were also performed using Galaxy; the FGSEA (fast preranked GSEA) was used for GSEA analysis. Enrichment analyses were carried out using Metascape (https://metascape.org/gp/index.html#/main/step1, accessed on 1 September 2020) [34].

### 2.8. RNA Sequencing Analysis

Cox proportional hazards (Cox PH) regression analyses were performed using the R survival package. The PH assumption was tested. Cutoff points were estimated using Maximally Selected Rank Statistics (the Maxstat package, https://cran.r-project.org/web/packages/maxstat/maxstat.pdf, accessed on 8 August 2020). The TCGA PanCancer Atlas pRCC dataset available from cBioPortal [35,36] was used.

### 2.9. Examination of Gene Expression

Gene expressions were determined using the UALCAN platform (ualcan.path.uab.edu/home, accessed on 31 March 2021) [37].

### 2.10. Statistical Analysis

Kaplan-Meier survival analyses and logrank test were conducted by R *Survival* package and tools provided by cBioPortal. Cox regression analyses were performed using R *survival* package. Time-dependent receiver operating characteristic (tROC) analyses were carried out with R *timeROC* package. ROC and precision-recall (PR) profiles were constructed using the PRROC package in R. Two-tailed Student *t*-test, one-way ANOVA, and two-way ANOVA were performed for statistical analysis of two and more than two groups respectively, with *p* < 0.05 to be considered statistically significant. Tukey’s test was performed for post-hoc analysis. Statistical analysis was conducted by GraphPad Prism 7 and data were presented as mean ± SEM/SD. A value of *p* < 0.05 was considered statistically significant.

## 3. Results

### 3.1. Association of OIP5 Upregulation with pRCC Tumorigenesis and Progression

OIP5 was reported to be a component gene in a multigene set predicting the risk of prostate cancer recurrence [38]; its upregulation associates with adverse features in ccRCC and bladder cancer [19,29], supporting a general involvement of OIP5 in urogenital cancers. To investigate this possibility, we examined OIP5 expression in pRCC using a tissue microarray (TMA) containing 40 pairs of pRCC and 74 pairs of ccRCC tumors with the adjacent non-tumor kidney (AJK) tissues from 20 and 37 patients, respectively. The pRCC patient population (*n* = 20) consists of 11 men and 9 women with most tumors being at T1 stage (Table 1). In comparison to AJK tissues, pRCC tumor tissues expressed a significant OIP5 upregulation (Figure 1A,B). OIP5 expression was further increased in advanced T stage tumors (Figure 1B). Consistent with a previous report, OIP5 upregulation occurred in Grade 2–3 ccRCC tumors compared to the AJK tissues; nonetheless, we could not demonstrate OIP5 upregulation in Grade 1 ccRCC compared to the AJK tissues (Appendix A), suggesting a role of OIP5 in ccRCC progression. By using the TCGA RNA-sequencing data organized by UALCAN (ualcan.path.uab.edu/home, 31 March 2021) [37], OIP5 upregulation at the mRNA level in pRCC tissues was observed (Figure 1c); the upregulations reflects the level of severity and the order of unfavorable outcome of pRCC with higher expression levels in T2P over T1P tumors, CIMP tumors over other subtypes (Figure 1D), stage 3 tumors over stages 1–2 tumors, stage 4 over stage 3 tumors (Figure 1E), and N1 (lymph node metastasis) over N0 tumors (Figure 1F). Consistent with its associations with adverse tumor features, OIP5 expression robustly stratifies pRCC tumors into a high- and low-risk group based on overall survival possibility (Figure 1G). Among the 10 patients in the OIP5-high group, seven died in a rapid time course (Figure 1G). Collectively, these observations support a strong association of OIP5 with pRCC tumorigenesis and progression.

### 3.2. OIP5-Mediated Enhancement of pRCC Tumorigenesis along with Network Alterations

Attributed to the uncommon status of pRCC, there are only limited number of confirmed pRCC cell lines available. ACHN is the most widely used and confirmed metastatic pRCC cell line; the cells have the typical feature of *c-MET* polymorphism detected in pRCC [39,40]. ACHN is likely the only confirmed metastatic pRCC cell line [39]. To analyze the functional impact of OIP5 on pRCC tumorigenesis, we stably expressed OIP5 in ACHN cells (Figure 2A). In comparison to ACHN EV (empty vector) cells, ACHN OIP5 cells displayed elevated abilities for proliferation (Figure 2B), colony formation (Figure 2C; Appendix A), invasion (Figure 2D; Appendix A), and growth in soft agar (Figure 2E; Appendix A). We have also established the EV and OIP5 stable lines in the commonly used 786-O ccRCC cells, and OIP5 overexpression did not affect all of the above oncogenic events observed in ACHN cells in vitro (data not shown), which suggests a certain level of specificity of OIP5 in promoting pRCC. In vivo, OIP5 enhanced the growth of ACHN cell-produced xenografts compared to tumors produced by ACHN EV cells (Figure 2F); mice bearing ACHN OIP5 tumors reached endpoint faster compared to animals with ACHN EV cell-produced tumors (Figure 2G). The overexpression of OIP5 in ACHN OIP5 tumors was confirmed (Appendix A). The ACHN OIP5 tumors show a significant increase of CDK2 expression largely in the nuclei (Appendix A); the functions of this are not clear as no upregulations of the relevant cyclins (cyclin A and cyclin E) was observed (data not shown).

To further analyze factors and networks utilized by OIP5 in enhancing ACHN cell-produced xenografts, RNA-sequencing (RNA-seq) was performed on ACHN EV and ACHN OIP5 tumors at three per group. Gene set enrichment analysis (GSEA) was conducted on differentially expressed genes obtained in the setting of OIP5 vs. EV. When enrichment in the oncogenic gene sets (C6) collection was analyzed using FGSEA (fast gene set enrichment analysis), we observed that genes downregulated (DN) in cells with activation (UP) of ERB2, MEK, and mTOR were also downregulated in ACHN OIP5 tumors compared to ACHN EV tumors (Figure 3A), suggesting OIP5 suppressing those genes that are downregulated by ERB2, MEK, and mTOR. Similarly, ACHN OIP5 tumors also display downregulation of EGFR-downregulated genes (Appendix A). The serine/threonine kinase 33 (STK33) is a synthetic lethal interacting protein of KRAS mutant, i.e., cells expressing KRAS mutant rely on STK33 for survival [41]. Knockdown of STK33 in acute myeloid leukemia cells led to upregulation of a set of genes (STK33-UP) [41], suggesting a potential inhibition of these genes by STK33. These gene expressions were also reduced in ACHN OIP5 tumors (Figure 3A; Appendix A). To test the reliability of the enrichment obtained by FGSEA, GSEA was further conducted using a more stringent platform: EGSEA. Ensemble gene set enrichment analysis produces a consensus gene set ranking (enrichment) with the combination of multiple (up to *n* = 12) algorithms [42]. With the maximal stringent condition using all 12 algorithms, EGSEA revealed within the top 12 ranks the downregulation of the ERB2- and MEK-suppressed gene sets in ACHN OIP5 tumors (Appendix A); downregulation of genes in cells with STK33 knockdown was observed in multiple setting (Appendix A) which is consistent with the enrichments derived from using FGSEA (Appendix A). All top 12 ranked gene sets obtained by EGSA (Appendix A) are also included in those produced by FGSEA (Appendix A). It is intriguing that VEGFA-suppressed genes in HUVEC (human umbilical vein endothelial cell) cells were also downregulated in ACHN OIP5 tumors (Appendix A). Based on the overall gene set enrichment within the oncogenic gene set (C6, MSigDB) collection (Appendix A), we can summarize that in ACHN OIP5 xenografts, the RB pathway is inhibited and the signaling processes of STK33, BMI1, EZH2, MYC, WNT, VEGFA, and EGFR/ERB2 are enhanced (Figure 3B).

We further examined gene set enrichment within the Hallmark gene set collection using FGSEA. The analyses revealed downregulations of inflammatory response, TNFα_via_NFκB signaling (NFκB-regulated genes in response to TNFα), and complement gene expression (Hallmark_Component, normalized enrichment score/NES: −1.48, padj 0.013) (Appendix A). Additionally, ACHN OIP5 xenografts exhibited upregulations in gene sets regulating fatty acid metabolism and cholesterol homeostasis (Figure 3C; Appendix A). These enrichments were also produced by EGSEA (Appendix A). Several processes are enhanced in ACHN OIP5 tumors, which include oxidative phosphorylation, the expression of E2F and MYC targets, EMT (epithelial mesenchymal transition), mTORC1 signaling, and adipogenesis (Appendix A). Enrichment in glycolysis in ACHN OIP5 tumors was obtained by FGSEA (Appendix A), which was also confirmed by KEGG pathway analysis using EGSEA (Appendix A). Evidence thus suggests a metabolic switch to Warburg metabolism in ACHN OIP5 tumors.

### 3.3. Association of OIP5-Related Differentially Expressed Genes with CIMP Subtype

In comparison to other pRCC subtypes, CIMP tumors have a Warburg metabolic shift [12], indicating an association between OIP5-affected genes and CIMP. This notion is supported by the elevation of OIP5 expression in CIPM pRCC tumors (Figure 1D). To investigate this possibility, we firstly defined the differentially expressed genes (DEGs) in ACHN OIP5 tumors vs. ACHN EV tumors as those with *p*.adj < 0.05 and fold change > |1.5|; a total of 1128 DEGs were derived (Appendix A). In these DEGs, the top upregulated genes include WNT7A, FGF1, CNTN1 [43], SOX2, and others, which are known for their facilitative roles in tumorigenesis. The top 20 clusters enriched in these DEGs contain those that regulate urogenital system development, blood vessel morphogenesis, hippo pathway, cell surface receptor signaling, pathway in cancer, epithelial cell proliferation, and others (Figure 4A; Appendix A). Individual terms in these enriched clusters form a network connection (Appendix A). These pathways are clearly relevant to tumorigenesis. DEGs are clustered in ACHN OIP5 tumors vs. ACHN EV tumors (Figure 4B).

To confirm the relevance of these DEGs derived from ACHN cell-produced xenografts in pRCC pathogenesis, we analyzed their relationship to DEGs derived from primary pRCCs relative to OIP5 expression. In the TCGA Pancancer pRCC dataset within cBioPortal, high OIP5 expression robustly separates pRCC tumors into a high and low risk group based on their overall survival (OS) possibilities (Figure 1G). From these two groups, we obtained 873 DEGs defined by *q* < 0.05 and fold change ≥ |2| (Appendix A). These primary pRCC-derived DEGs share 66 overlap DEGs (Overlap66) with the xenograft-derived DEGs (Table 2; Figure 4C). The alterations in their expressions in normal kidney tissues (*n* = 30) and pRCC tumors (*n* = 290) at different stages are presented in Appendix A. The genes with further elevations in Stage 3–4 tumors include SLC7A11, PCSK5, STC2, PLK1, TK1, TRIB3, and SRXN1 (Appendix A).

Among these 66 DEGs, 8 and 41 genes are not known for associations with cancer and ccRCC respectively (Appendix A); only PLK1 was reported to be a component gene in a prognostic multigene of pRCC (Appendix A). Overlap66 is novel to pRCC. Forty-six out of 66 DEGs significantly predict overall survival (OS) possibility with some being individually efficacious based on their *p* values: 6.73 × 10^−10^ for PCSK5, 1.3 × 10^−10^ for C116orf75 (RMI2), 1.84 × 10^−13^ for ATAD2, and others (Appendix A). Furthermore, 33 DEGs retain their predictive significance after adjusting for age at diagnosis, sex, and T stage (Appendix A).

The potentials of the 33 DEGs as prognostic biomarkers are in accordance with their expression status in CIMP. C116orf75, SRXN1, TK1, and TRIB3 positively predict poor OS (Table 2; Appendix A); they are notably upregulated in CIMP tumors (Figure 4D). In reverse, CCDC106, CX3CL1, LYNX1, and SPATA18 are negatively associated with poor OS (Table 2; Appendix A); their expressions are particularly downregulated in CIMP tumors (Figure 4E). In all 46 genes with their expression associated with OS shortening, 11 show no alterations in gene expression in CIMP tumors (Table 2); for the remaining 35 genes, their positive and negative predictions of OS shortening correlate with their respective upregulation and downregulation in CIMP tumors (Table 2). This correlation of expression was not observed in tumors vs. non-tumor tissues (Table 2). In view of CIMP tumors having the poorest OS possibility [12], the association of these gene expression with CIMP tumors supports their potential as prognostic biomarkers.

### 3.4. Robust Prognostic Biomarker Potential of Overlap66 and Its Sub-Multigene Panels

Following the above analyses, we examined the OS-related prognostic potential of Overlap66 as a multigene panel. The expression data for these DEGs along with the relevant clinical data were retrieved from the Pancancer pRCC dataset within cBioPortal. Risk scores for individual tumors were calculated as ∑(coef_i_ × Gene_iexp_)_n_ (coef_i_: Cox coefficient of gene_i_, Gene_iexp_: expression of Gene_i_, *n* = 66). Coefs were obtained using the multivariate Cox model. Overlap66 risk scores efficiently predict OS shortening using both univariate (UV) and multivariate (MV) Cox models (Appendix A). The MV model consists of the risk scores, age at diagnosis, sex, and T stage (Appendix A). With the cutoff point optimized using the Maximally Selected Rank Statistics (Appendix A), Overlap66 effectively stratifies the risk of fatality (possibility of OS) and relapse (progression-free survival/PFS) (Figure 5A,B). The discriminations of OS and PFS are with time-dependent area under the curve (tAUC) value of 94.6–91.3% in the time frame of 12.4 month (M) to 57.2 M (Figure 5A) and 93.7–86.7% for 10.8 M to 50.6 M (Figure 5B), respectively. Collective evidence supports Overlap66 being a novel and robust prognostic multigene panel for pRCC.

We further validated Overlap66 risk score in stratification of pRCC fatality risk using a recently developed R package: contpointr (https://github.com/thie1e/cutpointr, accessed on 1 May 2021). An optimal cutoff point was obtained with Kernel smoothing model coupled with 1000 bootstrapping. This cutoff point classifies pRCC fatality risk at 0.78 sensitivity and 0.84 specificity or the sum of sensitivity and specificity (sum_sens_spec) value of 1.62 (Figure 6A). Risk stratifications of out-of-bag bootstrap samples (*n* = 1000) occurred most frequently at sum_sens_spec 1.6 (Figure 6B), which closely approximates sum_sens_spec 1.62 associated with the optimal cutoff point on the full cohort (Figure 6A). The fatality risk stratifications of the in-bag samples (*n* = 1000, average 63.2% of full samples) and the out-of-bag samples (*n* = 1000) were at the median sum_sens_spec values of 1.62 and 1.60 respectively. Taken together, these bootstrap analyses reveal a good out-of-sample performance of Overlap66 in classification of pRCC fatality risk, supporting Overlap66’s application in real world. This potential is strengthened by the effectiveness of the risk classification with a range of cutoff points (Figure 6B,C).

We subsequently optimized Overlap66. As OIP5 expression was at 1.9 folds in ACHN OIP5 tumors compared to ACHN EV tumors (Appendix A), we defined a subgroup of DEGs as those with *p*.adj < 0.05 and fold ≥ |1.9| in ACHN OIP5 tumors compared to ACHN EV tumors. These DEGs (*n* = 298) share 21 overlap genes (Overlap21) with primary pRCC-derived DEGs (Appendix A). As expected, Overlap21 is a subgroup of Overlap66 (Table 2). Overlap21 risk scores predict OS possibility under both UV and MV Cox models with comparable efficiency as Overlap66, evident by Hazard ratio (HR) and 95% confident interval (CI) (Appendix A). Similar prediction efficiencies for PFS between Overlap21 and Overlap66 were also observed (Appendix A). Overlap21 effectively stratifies the risk of mortality and PFS; the discriminations possess high tAUC values (Figure 5A,B). In comparison, Overlap21 seems marginally less effective compared to Overlap66 in the discriminations of OS and PFS (Figure 5A,B). Nonetheless, the Overlap21-mediated predictions are clearly effective. Similar to Overlap66, Overlap 21 risk score is an independent predictor of poor OS after adjusting age at diagnosis, sex, and T stage (Table 3).

The utility of Overlap21 in assessing pRCC fatality risk is further illustrated by its impressive separation of disease-specific survival (DSS) risk (Figure 7A,C). DSS is more specific compared to OS in addressing factors contributing to cancer-caused deaths. Overlap66 did not perform well in DSS estimation (data not shown), which might be attributable to the small number of events (disease-specific death *n* = 27) in the context of the large number of variables (*n* = 66 in Overlap66). We thus generated Overlap21plus by using Overlap21 as the basis, and the rest of DEGs within Overlap66 were added if they remain risk factors for decreased OS after adjusting age at diagnosis, sex, and T stages (Appendix A). However, Overlap21plus was not superior to Overlap21 in the estimation of OS and PFS (data not shown). Nonetheless, the risk score of Overlap21plus predicts DSS risk in a comparable efficiency as Overlap21 (Appendix A); its ability to classify DSS possibility was marginally superior to Overlap21 (Figure 7A–C).

Instead of using time-dependent ROC (receiver-operating characteristic) in evaluating the performance of Overlap66, Overlap21, and Overlap21plus for their prognostic prediction, we further examined their prediction performance using the intact population (i.e., without the time component) by both ROC-AUC and PR-AUC curves. The precision-recall (PR) curve is used to account for the imbalance nature of dataset; the event rates are 14.6% (41/280) for OS, 18.9% (53/280) for PFS, and 9.6% (27/280) for DSS, which are much less than 50%. PR-curve was suggested to evaluate biomarker’s discriminative performance [44]. According to both ROC-AUC and PR-AUC curves, Overlap66 predicts OS and PFS possibilities better than Overlap21 (Figure 5C,D), while Overlap21plus holds a slight edge over Overlap21 in estimating DSS possibility (Figure 7D,E).

### 3.5. Alterations in Immune Cell Subsets in High-Risk pRCC Tumors

Tumor-associated immune cells play critical role in tumor initiation and progression [45,46], suggesting alterations of immune components in Overlap66-stratified high-risk pRCC tumors compared to those of low-risk. To examine this possibility, we profiled all 22 leukocyte subsets in 280 primary pRCC tumors within the TCGA Pancancer dataset using CIBERSORTx (https://cibersortx.stanford.edu/index.php, accessed on 21 July 2021) [47]. Significant alterations in several immune cell subsets between high-risk (*n* = 32) and low-risk tumors (*n* = 248) were detected (Figure 8). Increases in B naïve cells, T follicular helper cells (Tfh), CD4 T memory (activated) cells, and CD8 T (*p* = 0.075) cells were detected in high-risk local pRCC tumors (Figure 8A), indicating persistent immune reactions towards tumors; this scenario is not uncommon, evident by the co-existence of ATM-derived tumor surveillance (antioncogenic actions) with oncogenic actions during cancer initiation and progression [48]. However, CD8 T cells expressed an upregulation of programmed cell death protein 1 (PDCD1 or PD1) (Figure 8B), a major mechanism contributing to CD8 T cell exhaustion in cancer [49]. Additionally, T regulatory (Treg) cells suppress T cells activation via downregulation of CD80/86 in antigen-presenting dendritic cells [50] and a significant elevation of Treg cells was observed in high-risk pRCC tumors (Figure 8A). Alterations in M1 and M2 composition in high-risk pRCCs (Figure 8A) are consistent with the contributions of tumor-associated macrophages in cancer progression [51]. Decreases in macrophages M2 in high risk pRCC tumors is supported by a downregulation of β-2-adrenergic receptor (ADRB2) in these tumors (Figure 8C); the receptor was associated with M2 macrophages [52]. Reductions of activated mast cells in high-risk pRCC tumors (Figure 8A) suggest a downregulation of immune reactions in facilitating pRCC progression. While B naïve cells, CD8 T cells, M2 macrophages, and activated master cells are similarly clustered in both Overlap66 stratified high- and low-risk pRCCs (Appendix A), activated CD4 T memory cells, Tfh, Treg, and M1 macrophages in the high-risk tumors display different clustering patterns from their counterparts in the low-risk pRCCs (Figure 8D–G). Collectively, changes in immune components in high-risk pRCC tumors stratified by Overlap66 risk scores favor the development of an immune suppressive microenvironment, which might be a mechanism underpinning pRCC progression. This concept provides additional evidence supporting Overlap66 being a novel and effective prognostic biomarker for pRCC.

### 3.6. Critical Contributions of PLK1 to OIP5-Promoted Growth of pRCC Tumors

PLK1 (Polo-like kinase 1) is one of the upregulated DEGs identified in relation to OIP5 upregulation in both xenograft tumors and primary pRCC, i.e., a component gene of Overlap66 (Table 2). In the same manner, both LYPD6 and PCSK5 were upregulated in primary pRCC tumors with elevated OIP5 expression and in ACHN OIP5 xenografts determined by RNA-seq (Table 2). By using real-time PCR, we confirmed LYPD6 (fold 2.32 ± 0.2/SD, *p* < 0.5) and PCSK5 (fold 2.6 ± 0.1, *p* < 0.05) upregulations in ACHN OIP5 tumors (*n* = 4) compared to ACHN EV tumors (*n* = 6). PLK1 upregulation in xenografts produced by ACHN OIP5 cells compared to those derived from ACHN EV cells was demonstrated by RNA-seq and real-time PCR (Figure 9A,B)). In primary pRCC tumors, OIP5 expression correlates with PLK1 expression with a Pearson correlation value of 0.7 (UALCAN, ualcan.path.uab.edu/home, 1 March, 2021). OIP5, which is also known as Mis18β, is an essential component of the Mis18 complex that is required to load a histone H3 variant CENP-A (centromere protein A) to centromere of newly synthesized DNA strand in early G1 phase [53,54]. PLK1 contributes to CENP-A loading via phosphorylation of M18BP1, a component of the Mis18 complex [55]. In line with this knowledge, we examined whether PLK1 kinase activity plays a role in OIP5-promoted pRCC growth.

PLK1 inhibitors have been developed and approved by FDA as Orpha Drug Designation for cancer therapy [56,57]. The PLK1 inhibitor BI2536 caused G2/M arrest with concurrent reduction in G1 phase in ACHN OIP5 cells without apparent effects on cell cycle distributions of ACHN EV cells at the conditions used (Figure 9C). We then treated mice bearing ACHN EV or ACHN OIP5 cell-produced xenograft tumors with BI2536 when tumors reached 100 mm^3^. In the vehicle treatment group, the OIP5 tumors grew significantly faster compared to the EV tumor (Figure 9D). Administration of BI2536 had no effects on the growth of ACHN EV tumors but significantly inhibited the growth of ACHN OIP5 tumors (Figure 9E,F). In the presence of BI2536, ACHN OIP5 tumor showed marginally slower growth compared to ACHN EV tumors (Figure 9G). Inhibition of PLK1 significantly increases the survival of mice bearing ACHN OIP5 tumor (Figure 9H). As ACHN is a metastatic pRCC cell line [39], evidence supports inhibition of PLK1 being an option in treating metastatic pRCCs with OIP5 upregulation. Collectively, the above observations indicate synthetic lethality between OIP5 and PLK1 in metastatic pRCCs.

## 4. Discussion

Papillary RCC is a minor type of RCC compared to ccRCC which composes 75–80% of RCC cases. Nonetheless, pRCC can be as aggressive as ccRCC, particularly the T2P tumors which usually have more aggressive potential than ccRCC. As a minor RCC type, research on pRCC falls short compared to ccRCC. Therefore, the current understanding on pRCC remains limited, which presents a major concern particularly considering pRCC being associated with poor prognosis. The situation calls for improvement in risk assessment and personalized therapies in managing pRCC.

We provide here the first evidence for OIP5 being an important oncogenic factor of pRCC. This concept is supported by multiple pieces of evidence with respect to the impact of OIP5 upregulation on the tumorigenesis of pRCC cells in vitro and in vivo as well as the association of OIP5 upregulation with primary pRCC. Although we have made extensive efforts to knockdown OIP5 in ACHN cells, the attempts were not successful, suggesting OIP5 being essential for ACHN cell survival. This plausibility is in accordance with OIP5 initiating multiple processes critical for pRCC tumorigenesis, including those regulating urogenital system development, immune reaction, and others. Among these features is the expression status of OIP5 and its related DEGs within Overlap66 in CIMP. Although it remains to be determined whether OIP5 and these DEGs contribute to CIMP, this possibility seems likely. Among the pRCC subtypes, CIPM tumors are associated with a metabolic shift towards Warburg metabolism, which include enhancement of glycolysis, fatty acid and lipid metabolism, and hypoxia [12]. These are typical pathways enriched in ACHN OIP5 tumors (Figure 3; Appendix A).

OIP5 may also utilize other pathways in promoting pRCC. As an essential component (Mis18β) in the Mis18 complex, OIP5 is required for CENP-A loading and thus centromere formation [53,54]. This process is essential for genome stability, evident by the centromere-mediated chromosome segregation. In line with this concept, genes with function in maintaining genome stability are overrepresented in Overlap66; RMI2 (RecQmediated genome instability 2; C16ORF75) [58], RAD54B (RAD54 homolog B) [59], and PLK1 [60] all play roles in genome stability. Furthermore, pathway enrichment analysis of Overlap66 DEGs revealed the top pathways enriched being GO:0071168: protein localization to chromatin (*p* < 0.0001), GO:0140013: meiotic nuclear division (*p* < 0.0001), GO:0006790: sulfur compound metabolic process (*p* < 0.001), and GO:0000724: double-strand break repair via homologous recombination (*p* < 0.001).

One of the neighboring genes to OIP5 is OIP5-AS1 (OIP5 antisense RNA 1). According to GRCh38.p13 (Genome Reference Consortium Human Build 38 patch release 13) released in Feb 28, 2019, the OIP5 genes runs from 41,332,591 to 41,309,273 on chromosome 15 (https://www.ncbi.nlm.nih.gov/gene/11339, accessed on 29 August 2021), while the OIP5-AS1 gene runs from 41,282,697 to 41,313,338 on chromosome 15 (https://www.ncbi.nlm.nih.gov/gene/729082, accessed on 29 August 2021). While both genes have an overlap region of 4065 nucleotides, there is no evidence suggesting a regulatory relationship between OIP5 and OIP5-AS1 [61]. OIP5-AS1 encodes a long non-coding RNA (lnRNA) and possesses oncogenic activities via regulating a set of microRNAs [61]. For instance, OIP5-AS1 was reported to sponge miR-143-3P to enhance cervical cancer [62] and miR-186a-5p to facilitate hepatoblastoma [63]. However, the involvement of OIP5-AS1 in pRCC remains unknown. In view of both OIP5 and OIP5-AS1 being pro-oncogenesis and their adjacent genetic locations, potential functional connections between both in pRCC pathogenesis and progression is worthy of future investigation. In supporting this possibility, we noticed OIP5-AS1 being upregulated (1.37 folds, *p* = 0.00464 and *q* = 0.0459) in pRCC tumors expressing high levels of OIP5 compared to those with low levels of OIP5 expression (Figure 1G).

OIP5 is a tumor-associated antigen (TAA), owning to its largely restricted expression in human testis and its upregulation in multiple cancers [64,65]. We noticed that testis-associated proteins are also enriched in Overlap66, including OIP5, TEX15 (testis expressed 15, meiosis, and synapsis associated), SPAG1 (sperm associated antigen 1), and SPATA18 (spermatogenesis associated 18) (Table 2). It is thus tempting to propose an involvement of some testis events in pRCC tumorigenesis. OIP5 possesses a robust prognostic potential (Figure 1G). This predictive power is significantly strengthened in OIP5-derived multigene sets: Overlap66, Overlap21, and Overlap21plus. Because of the small number of component genes and its effectiveness in predicting OS, PFS, and DSS, Overlap21 may offer primary clinical application with the other two provide assisting roles. These multigene sets possess great potential to be implemented into clinical applications. This possibility is supported by a very good out-of-sample performance of Overlap66 in stratification of pRCC fatality risk (Figure 6) and these stratifications can be effective using a range of cutoff points (Figure 6B,C). Clinical applications of Overlap66 and its-related multigene panels may significantly improve our ability in predicting prognosis and potentially even the development of personalized therapies.

Although a recent phase 2 clinical trial suggests the MET inhibitor cabozantinib improves PFS and OS in patients with metastatic pRCC compared to a current standard of care with sunitinib [66], much more needs to be done to confirm its efficacy. The dependence on MET signaling is likely much less in T2P compared to T1P, which needs to be considered in using MET inhibitors in treating patients with T2P tumors. Our finding of PLK1 inhibitor being effective in inhibition of ACHN OIP5 tumor growth may have significant clinical applications in treating metastatic pRCC with OIP5 upregulation; this will offer a venue for potential utilization of personalized therapy in pRCC. This possibility can be readily explored as volasertib, a PLK1 inhibitor, has been granted Orpha Drug Designation status in treating AML (acute myeloid leukemia) in 2014 and rhabdomyosarcoma in 2020 (https://oncoheroes.com/press-releases-content/2020/10/14/volasertib-a-potential-new-treatment-for-rhabdomyosarcoma-receives-orphan-drug-designation-from-the-us-fda, accessed on 31 May 2021) by FDA. Even for BI2356 used in this study, its clinical safety was deemed acceptable based on multiple phase II clinical trials (NCT00701766, NCT00376623, and NCT00526149) on solid cancer. Intriguingly, we observed changes in immune cells in pRCC tumors stratified by Overlap66, an OIP5-derived multigene panel, including increases of Treg cells and PD1 upregulation in CD8 T cells (Figure 8). This suggests that these patients might benefit from rescuing of CD8 T cell exhaustion via PD1-based immune therapies. Treg action can be suppressed via CTLA-4 immune therapy. In this regard, combinations of PLK1 inhibitor and PD1 or CTLA-4 immune therapies might optimize personalized treatment. Collectively, this research enhances our understanding of pRCC and suggests novel means in predicting pRCC prognosis and in developing personalized therapy. Nonetheless, additional work is required to realize these potentials.

## 5. Conclusions

We report here a novel and thorough investigation of OIP5’s contributions to pRCC. OIP5 upregulations robustly predict the survival possibility of pRCC patients. The multigene panel Overlap66, a portion of the OIP5 network, possesses an impressive prognostic potential in predicting pRCC progression, disease-specific survival, and overall survival; the predictions are associated with an excellent out-of-sample performance, indicating its potential clinical applications. Furthermore, PLK1 is among Overlap66 and displays synthetic lethality with OIP5; inhibition of PLK1 using BI2356 only suppresses the growth of xenograft tumors generated by ACHN OIP5 cells but not the growth of tumor produced by ACHN EV cells, supporting a targeted and personalized therapy for pRCCs with OIP5 elevations. Collectively, combinational use of Overlap66 and PLK1 inhibitors may open an era of personalized therapy in pRCC.

## 6. Patents

A US provisional patent (63/202,616) has been filed.

## Figures and Tables

**Figure 1 cancers-13-04483-f001:**
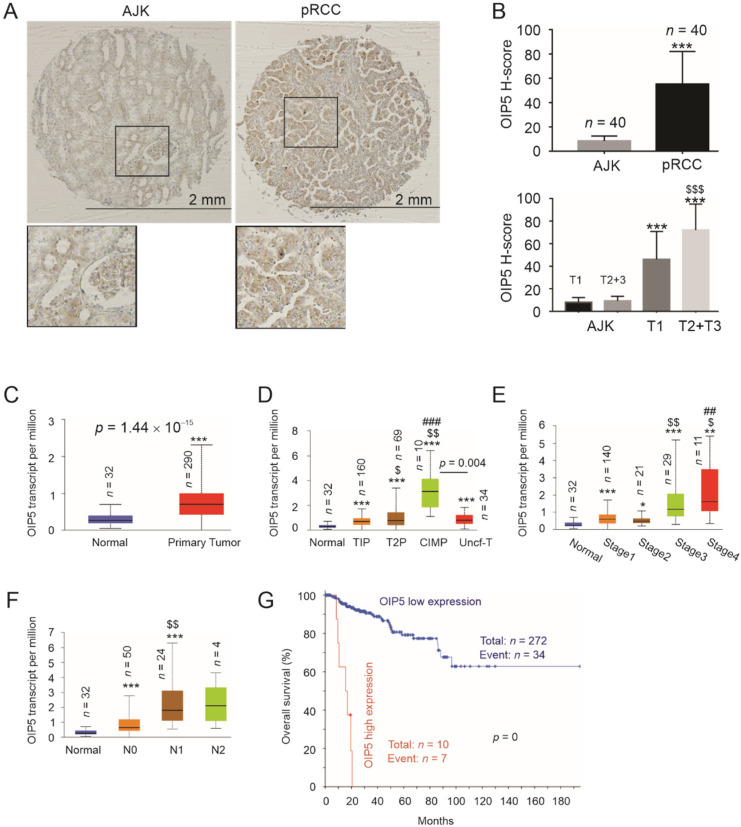
Upregulation of OIP5 associates with adverse features of pRCC and predicts poor overall survival. (**A**) IHC staining for OIP5 was performed using a RCC TMA; typical images of OIP5 staining in the adjacent kidney (AJK) and pRCC tumor tissues are presented. (**B**) Quantification of OIP5 IHC staining by H-score in the indicated tissues; means ± standard deviations (SDs) are graphed. Statistical analyses were performed using 2-tailed Student’s *t*-test; ***: *p* < 0.001 compared to the respective AJK tissues, $$$: *p* < 0.001 compared to T1 tumors. (**C**–**F**) OIP5 mRNA expressions in the indicated setting were analyzed using the TCGA dataset organized by UALCAN [37]. Student’s *t*-test (**C**) and other indicated paired statistics were provided by UALCAN. *: *p* < 0.05, **: *p* < 0.01, ***: *p* < 0.001 compared to normal kidney tissues; $: *p* < 0.05, $$: *p* < 0.01 compared to T1P (**D**), Stage 2 (**E**), and N0 (**F**); ##: *p* < 0.01, ###: *p* < 0.001 compared to T2P (**D**) and Stage 3 (**E**). (**G**) Survival analysis was performed using the TCGA Pancancer pRCC dataset within cBioPortal. Logrank test was performed. Cutoff point used to separate the high- and low-OIP5 expression groups was ≥2 z-score or 2SD. The graph was produced using tools provided by cBioPortal. The median months overall survival for patients in the high-OIP5 group was 15.48 months.

**Figure 2 cancers-13-04483-f002:**
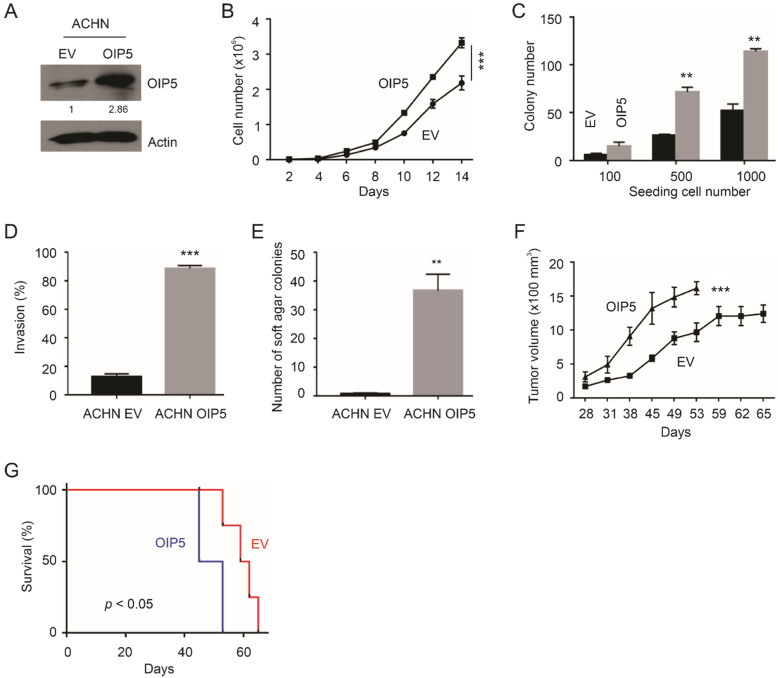
OIP5 promotes oncogenic processes of ACHN cells in vitro and in vivo. (**A**) ACHN empty vector (EV) and OIP5 stable lines. Western blot was carried out using anti-OIP5 and Actin antibodies. OIP5 expression was normalized to Actin and presented at fold changes to OIP5 expression in EV cells. (**B**) ACHN EV and ACHN OIP5 cells were seeded in 6-well plate at 10^5^ cell/well; cell numbers were recorded at the indicated days. Experiments were repeated three times; means ± SDs are graphed. Statistical analysis was performed using 2-way ANOVA. ***: *p* < 0.001 between the two curves. (**C**) The indicated cells were seeded at the indicated number in 6-well plates. Colonies were formed following 2 weeks culture. Experiments were repeated three times; means ± SDs are graphed. **: *p* < 0.01 compared to the respective EV by Student’s *t*-test (2-way). (**D**,**E**) Invasion and soft agar assays were repeated 3 times; means ± SDs are graphed. **: *p* < 0.01, ***: *p* < 0.001 compared to the respective EV control by Student’s *t*-test (2-way). (**F**,**G**) Xenografts were produced in NOS/SCID mice (5 mice per group) using ACHN EV cells and ACHN OIP5 cells. Means ± SEM (standard error of the mean) are graphed; ***: *p* < 0.001 between the two curved by two-way ANOVA (**F**). Kaplan-Meier curve; statistical analysis was performed using logrank test (**G**).

**Figure 3 cancers-13-04483-f003:**
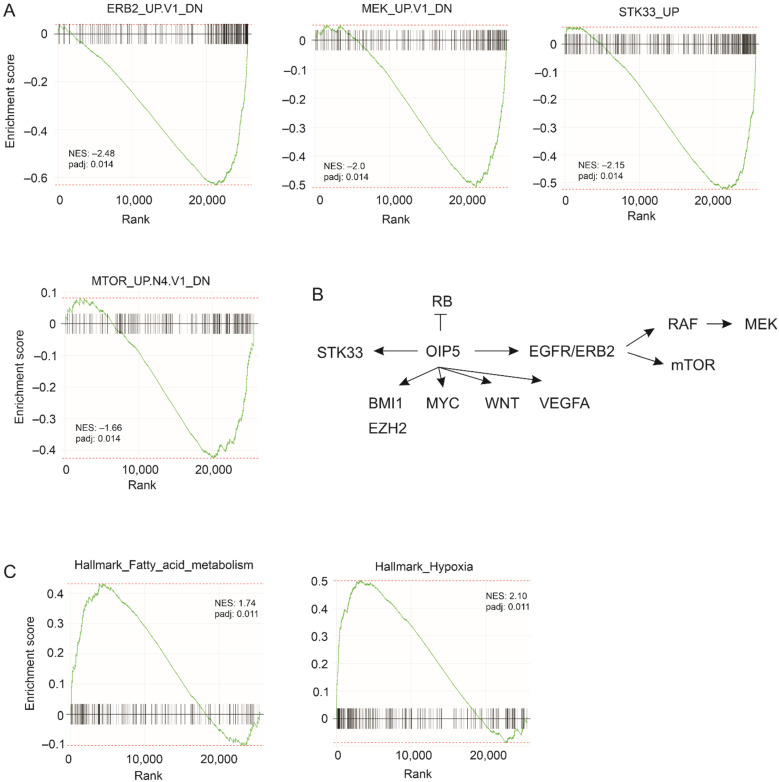
OIP5 induces network alterations during pRCC tumorigenesis. (**A**) GSEA (gene set enrichment analysis) on differentially expressed genes derived from the comparison of ACHN OIP5 tumors to ACHN EV tumors was performed with FGSEA within the Galaxy platform. The MSigDB oncogenic gene sets (C6) collection was used. (**B**) Summary of the major oncogenic gene sets affected in ACHN OIP5 tumors (see Appendix A for individual gene sets affected). (**C**) Enrichment of the indicated gene set within the MSigDB hallmark gene sets collection (see Appendix A for individual gene sets affected).

**Figure 4 cancers-13-04483-f004:**
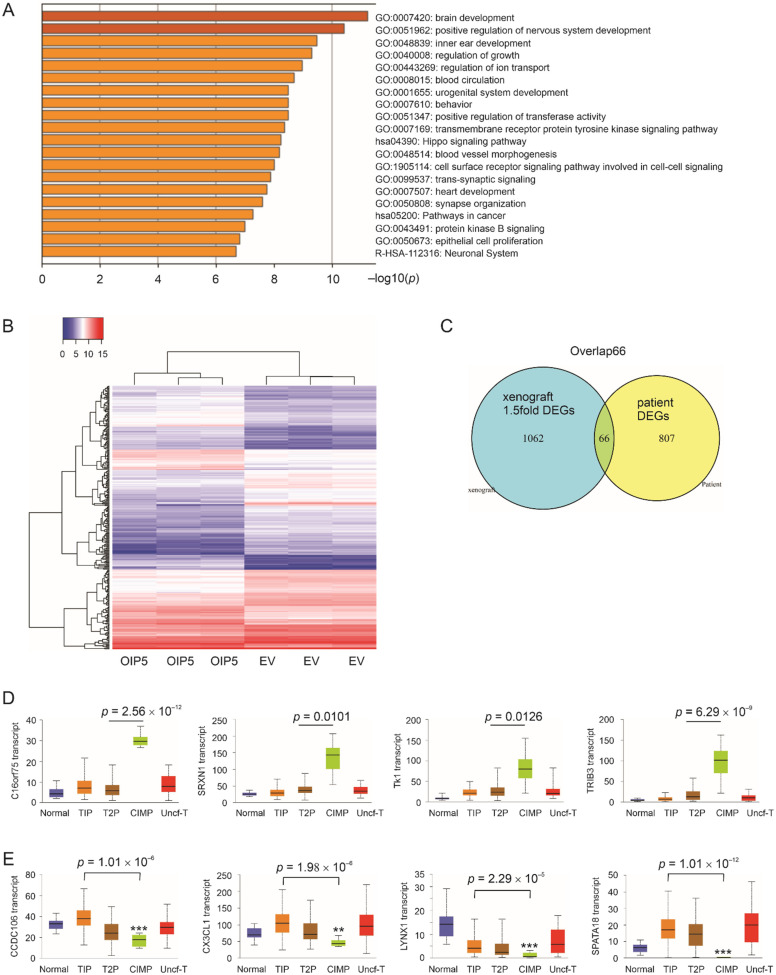
Pathway enrichment of OIP5 DEGs. DEGs were first defined as *p*.adj < 0.05 and fold changes > |1.5| in the comparison of ACHN OIP5 tumors (*n* = 3) vs. ACHN EV tumors (*n* = 3). (**A**) Pathway enrichment in these DEGs (Appendix A) was then performed using the Metascape [34] platform. (**B**) Clustering of DEGs in ACHN OIP5 tumors and ACHN EV tumors. (**C**) The number of overlapping genes between primary (patient) pRCC-derived DEGs and DEGs obtained from xenografts at fold change > |1.5|. (**D**,**E**) The indicated DEGs were analyzed for expression in the histological subtypes of pRCC using the UALCAN platform [37]. DEGs positively (**D**) and negatively (**E**) predict shortening of OS (see Table 2 for details). **: *p* < 0.01, ***: *p* < 0.001 in comparison to normal kidney tissues.

**Figure 5 cancers-13-04483-f005:**
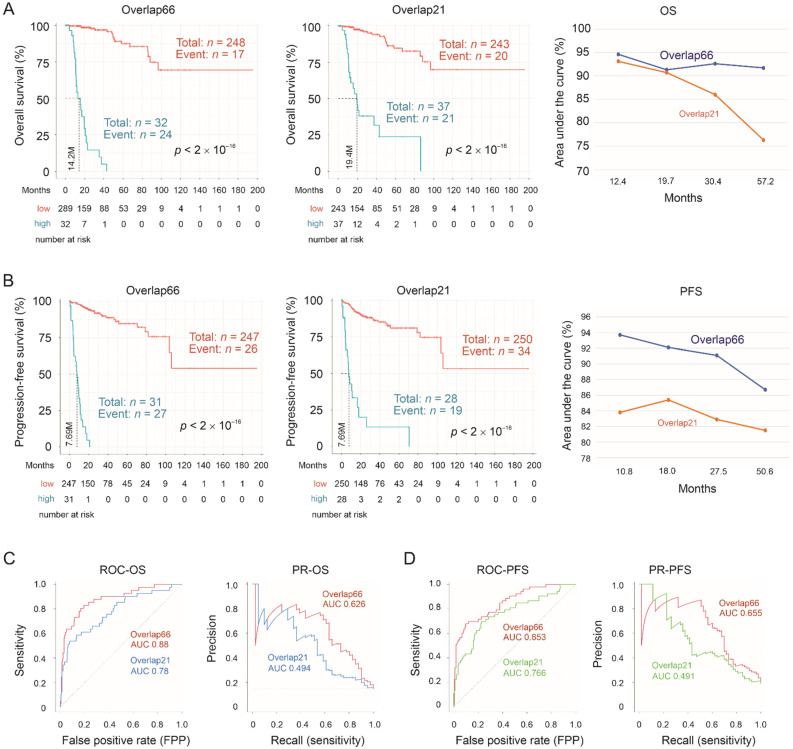
Stratification of the possibilities of overall survival (OS) and progression-free survival (PFS) by Overlap66 and Overlap21. (**A**,**B**) Cutoff points were determined by Maximally Selected Rank Statistics for the risk scores of Overlap66 (see Appendix A) and Overlap21. Kaplan Meier curves for OS (**A**) and PFS (**B**) are constructed, using the R survival package, with the populations at risk in the indicated follow-up period included. Statistical analyses were performed using logrank test. The median months of OS and PFS are indicated. Time-dependent ROC (receiver operating characteristic; tROC) curves were generated using the R *timeROC* package; time-dependent area under the curve (AUC) values for the indicated multigene sets are shown. (**C**,**D**) ROC and precision-recall (PR) curves for Overlap66 and Overlap21 in predicting OS and PFS possibilities were produced using the R PRROC.

**Figure 6 cancers-13-04483-f006:**
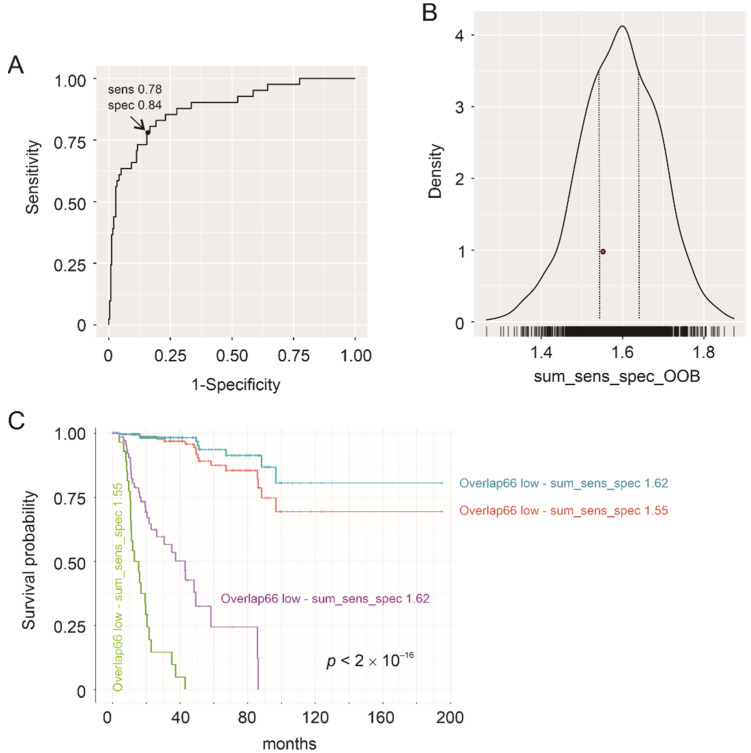
Validation of Overlap66 risk score in stratification of pRCC fatality risk. Cutoff points were estimated using Kernel smoothing method coupled with bootstrapping (*n* = 1000). The average in-bag and out-of-bag (OOB) bootstrap samples are 63.2% and 36.8% of the full sample size respectively. The analysis was performed using the cutpointr R package (https://github.com/thie1e/cutpointr, accessed on 21 July 2021). (**A**) ROC curve with the optimal cutoff point indicated (arrow); sens: sensitivity, spec: specificity, and the sum_sens_spec: 1.62. (**B**) Distribution of out-of-bag (OOB) metric values. The most predictions occur in these OOB samples (*n* = 1000) at the sum_sens_sepc value 1.6. The region marked by the 2 dotted lines includes a range of sum_sens_sepc values that frequently stratify the fatality risk with high accuracy. The red dot represents a sum_sens_sepc value 1.55. (**C**) Classification of pRCC tumors into a high- and low-risk group using two indicated cutpoints; the sum_sens_sepc 1.62 cutoff point was obtained using Kernel method and the sum_sens_sepc 1.55 cutoff point (see the red dot in panel (**B**)) was derived using Maximally selected LogRank statistics (see Appendix A). The *p* value is for both separations.

**Figure 7 cancers-13-04483-f007:**
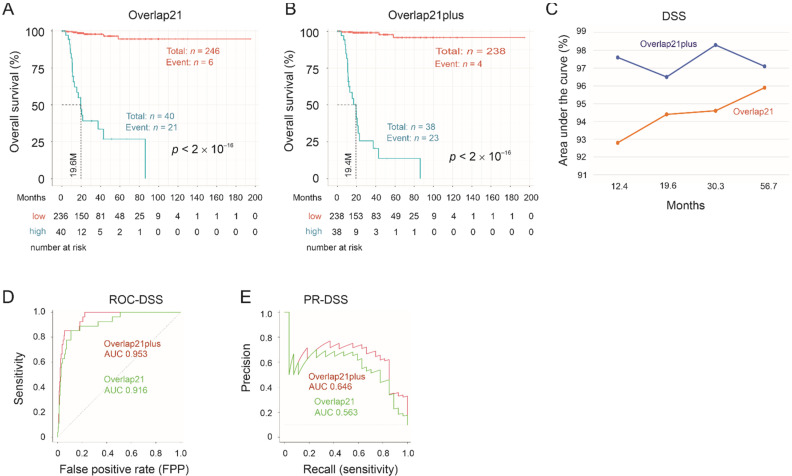
Stratification of the risk of disease-specific survival (DSS) by Overlap21 and Overlap21plus. (**A**,**B**) Separation of a high- and low-risk group based on DSS by Overlap21 and Overlap21plus risk scores. (**C**) tROC analysis. (**D**,**E**) ROC-AUC and PR-AUC curves for the indicated events.

**Figure 8 cancers-13-04483-f008:**
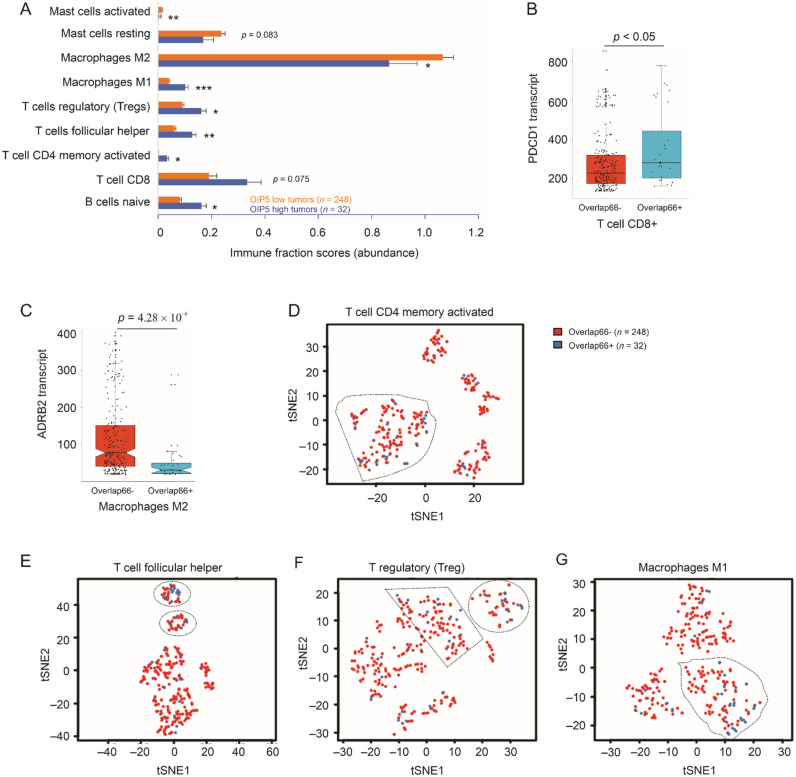
Changes in immune cells in pRCC tumors with high risk of fatality. RNA-seq profiles for 280 pRCC tumors were retrieved from cBioPortal and analyzed for immune cell profiles using the LM22 signature matrix and the CIBERSORTx program (https://cibersortx.stanford.edu/index.php, accessed on 21 July 2021) [47]. The analysis setting was with B-mode batch correction and 500 permutations (https://cibersortx.stanford.edu/index.php, accessed on 21 July 2021). (**A**) The abundance of the indicated immune cell subsets was determined by their immune fraction scores. Means ± SEMs in high-risk and low-risk tumors stratified by Overlap66 risk score are graphed; *: *p* < 0.05; **: *p* < 0.01; and ***: *p* < 0.001 in comparison to low-risk tumors by 2-tailed *t*-test. (**B**,**C**) Boxplots for the expression of PDCD1 and ADRB2 in low- and high-risk pRCC tumors; statistical analyses were conducted using Welch *t*-test with *p*-value adjusted with the Holm–Bonferroni (Holm) method. (**D**–**G**) Clustering of the indicated immune cell types associated with low risk (Overlap66−) and high risk (Overlap66+) tumors by tSNE (t-distributed stochastic neighbor embedding); the marked clusters are enriched with high-risk tumors. The graph was produced using CIBERSORTx (https://cibersortx.stanford.edu/index.php, accessed on 21 July 2021).

**Figure 9 cancers-13-04483-f009:**
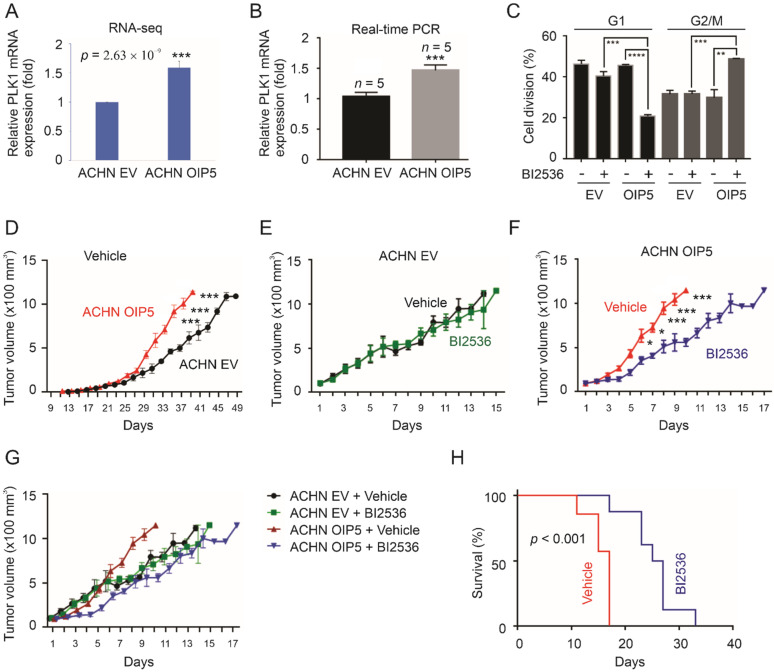
PLK1 inhibitor reduces OIP5-mediated pRCC tumorigenesis. (**A**,**B**) RNA-seq and real-time PCR analyses of PLK1 expression in ACHN EV and ACHN OIP5 tumors. RNA-seq was performed in 3 each from ACHN EV and ACHN OIP5 tumors. (**C**) ACHN EV and ACHN OIP5 cells were treated with DMSO (−) or the PLK1 inhibitor (BI2536) in 40 nM PLK1 inhibitor for 72 h, followed by quantification of cell cycle distributions. Experiments were repeated 3 times; means ± SEMs are graphed. **: *p* < 0.01, ***: *p* < 0.001, ****: *p* < 0.0001 in the indicated comparisons by 2-tailed Student’s test. (**D**–**G**) Mice bearing ACHN EV or ACHN OIP5 tumors were treated with vehicle or BI2536 (50 mg/kg) intravenously. The overall profiles of tumor growth in the vehicle treated setting (**D**); tumor volumes were recorded following treatments (**E**–**G**). Statistical analyses were performed using 2-tailed Student’s *t*-test; *: *p* < 0.05, ***: *p* < 0.001. (**H**) Kaplan Meier curve for the indicated mice reaching endpoints. Statistical analysis was performed using logrank test.

**Table 1 cancers-13-04483-t001:** The clinical parameters of pRCC patients included in TMA.

Parameter	Age (Year)	Male (*n*)	Female (*n*)	T1 (*n*)	T2 (*n*)	T3 (*n*)
Details	49.5 (39.8–61)	11	9	13	6	1

Age: median (Q1/quartile 1–Q3) *n*: number of cases. All patients were without lymph node metastasis (N0) and distant metastasis (M0).

**Table 2 cancers-13-04483-t002:** Characterization of Overlap66 genes.

Gene	OS ^1^	*p* Value	CIMP ^2^	*p* Value	Tumor ^3^	*p* Value
SLC7A11	+	0.0135 *	Up	8.4 × 10^−5^ ***	Up	1.68 × 10^−5^ ***
PCSK5 ^4^	+	6.73 × 10^−10^ ***	Up	0.0085 **	Down	0.011 *
STC2 ^4^	+	1.72 × 10^−6^ ***	Up	2.03 × 10^−5^ ***	None	NS
TEX15 ^4^	+	0.00101 **	Up	0.074	Down	1.55 × 10^−5^ ***
ESCO2 ^5^	+	6.89 × 10^−10^ ***	Up	0.0149 *	Up	1.67 × 10^−12^ ***
OIP5 ^4^	+	7.41 × 10^−12^ ***	Up	8.07 × 10^−5^ ***	Up	1.44 × 10^−15^ ***
PLK1 ^5^	+	5.58 × 10^−15^ ***	Up	0.0241 *	Up	<1 × 10^−12^ ***
ELOVL2 ^5^	+	4.36 × 10^−8^ ***	Up	0.0197 *	Up	4.34 × 10^−7^ ***
LYPD6 ^4^	N	NS	Up	0.0236 *	Down	4.82 × 10^−8^ ***
ATAD2 ^5^	+	1.84 × 10^−13^ ***	Up	0.00446 **	Up	3.80 × 10^−8^ ***
ISM1 ^4^	+	0.00237 **	N	NS	None	NS
TK1 ^5^	+	1.51 × 10^−10^ ***	Up	0.0126 *	Up	5.97 × 10^−8^ ***
TRIB3 ^5^	+	2.26 × 10^−6^ ***	Up	6.19 × 10^−9^ ***	Up	3.23 × 10^−13^ ***
KIAA1324L ^4^	N	NS	Up	0.0346 *	None	NS
SLIT3 ^5^	+	0.000677 ***	N	NS	Down	6.05 × 10^−9^ ***
COL14A1	+	0.00114 **	N	NS	Down	3.21 × 10^−7^ ***
FAM40B	N	NS	Up	0.0203 *	None	NS
STOX1	N	NS	N	NS	Down	1.89 × 10^−14^ ***
ABCA12 ^5^	+	9.87 × 10^−5^ ***	Up	0.0182 *	Up	<1 × 10^−12^ ***
RGS20	N	NS	Up	0.0377 *	Up	1.63 × 10^−12^ ***
ACCN2	+	0.0105 *	Up	1.57 × 10^−4^ ***	None	NS
DPYSL3 ^5^	+	2.18 × 10^−6^ ***	Up	3.37 × 10^−4^ ***	Down	7.92 × 10^−5^ ***
STAT4 ^5^	+	0.024 *	N	NS	Up	7.34 × 10^−8^ ***
CALCRL ^4^	+	0.0378 *	Up	0.0024 **	Down	7.29 × 10^−11^ ***
SRXN1	+	0.0258 *	Up	0.0102 *	Up	3.39 × 10^−10^ ***
FAR2	N	NS	Down^2a^	0.0049 **	Down	0.0165 *
TPD52 ^5^	+	8.63 × 10^−7^ ***	Up	8.08 × 10^−4^ ***	Down	1.62 × 10^−12^ ***
ZNF239 ^4^	+	8.96 × 10^−6^ ***	Up	0.00488 **	None	NS
C16orf75 ^5^	+	1.3 × 10^−10^ ***	Up	2.56 × 10^−12^ ***	Up	6.13 × 10^−10^ ***
HEYL ^5^	+	0.000855 ***	Up	0.0312 *	Down	2.52 × 10^−6^ ***
F2R ^4^	+	7.34 × 10^−7^ ***	Up	1.89 × 10^−5^ ***	Down	2.81 × 10^−9^ ***
KCNJ8 ^4^	+	0.000334 ***	N	NS	Down	4.02 × 10^−5^ ***
RAD54B ^5^	+	0.000931 ***	Up	0.0197 *	Up	1.14 × 10^−9^ ***
KCNK1 ^4^	+	4.35 × 10^−6^ ***	Up	0.00816 **	Down	0.00236 **
ZNF391 ^5^	+	0.00543 **	N	NS	Down	4.24 × 10^−4^ ***
POLR3G ^5^	+	0.000266 ***	N	NS	Up	1.74 × 10^−5^ ***
MEIS1	N	NS	Up	0.0046 **	Down	9.44 × 10^−12^ ***
MCM8 ^5^	+	0.00612 **	N	NS	None	NS
SNX16 ^5^	+	2.29 × 10^−7^ ***	Up	3.51 × 10^−5^ ***	None	NS
SPAG1 ^5^	+	0.000246 ***	Up	5.15 × 10^−4^ ***	None	NS
CX3CL1 ^5^	−	0.000679 ***	Down	1.98 × 10^−6^ ***	Up	4.47 × 10^−8^ ***
DYNC2LI1	−	0.00415 **	Down	2.01 × 10^−4^ ***	Up	1.62 × 10^−12^ ***
ACSS2 ^4^	−	0.0214 *	Down	1.62 × 10^−12^ ***	Down	1.73 × 10^−5^ ***
HS3ST5 ^4^	N	NS	Down	6.1 × 10^−5^ ***	None	NS
DPF3 ^4^	N	NS	Down ^2a^	0.0027 **	Down	3.98 × 10^−5^ ***
ZNF862	N	NS	Down	1.83 × 10^−11^ ***	Up	7.87 × 10^−12^ ***
LHPP ^5^	−	0.00907 **	N	NS	Down	0.0496 *
PITPNM3	−	0.0391 *	N	NS	Down	7.08 × 10^−11^ ***
GNG7 ^5^	−	0.000249 ***	N	NS	Down	3.30 × 10^−9^ ***
CHD5 ^4^	N	NS	Down	6.26 × 10^−5^ ***	None	NS
CCDC106 ^5^	−	0.000256 ***	Down	1.01 × 10^−6^ ***	None	NS
NBL1	-	0.0211 *	Down	4.47 × 10^−5^ ***	Up	<1 × 10^−12^ ***
LYNX1 ^5^	−	0.00675 **	Down	2.29 × 10^−5^ ***	Down	2.29 × 10^−8^ ***
PHYHIP	N	NS	Down	4.79 × 10^−4^ ***	None	NS
NRXN3	N	NS	N	NS	Down	1.87 × 10^−9^ ***
TMEM130 ^4^	N	NS	Down	2.25 × 10^−12^ ***	Down	4.59 × 10^−12^ ***
EREG ^4^	N	NS	Down	0.00318 **	Up	1.70 × 10^−12^ ***
C2orf62	−	0.00479 **	Down	1.97 × 10^−4^ ***	Up	1.62 × 10^−12^ ***
CCDC135	−	0.0478 *	Down	<1 × 10^−12^ ***	Up	1.62 × 10^−12^ ***
SYCE1L	N	NS	Down	2.56 × 10^−12^ ***	Up	<1 × 10^−12^ ***
GAL3ST3	N	NS	Down	1.63 × 10^−12^ ***	Down	7.38 × 10^−4^ ***
SPATA18 ^5^	−	1.82 × 10^−7^ ***	Down	1.62 × 10^−12^ ***	Up	1.62 × 10^−12^ ***
C6orf138 ^4^	N	NS	Down	0.026 *	Up	1.62 × 10^−12^ ***
ABI3BP	N	NS	Down	5.83 × 10^−11^ ***	Up	<1 × 10^−12^ ***
CNTN6 ^4^	N	NS	Down	5.07 × 10^−11^ ***	Up	<1 × 10^−12^ ***
SCEL ^4^	−	0.0331 *	Down	1.66 × 10^−12^ ***	Up	<1 × 10^−12^ ***

1: prediction of overall survival determined by univariate Cox analysis; +, −, and N: gene expression positively, negatively, and not predicting OS, respectively. NS: not significant. 2: expression status in CIMP tumors, “Up”: upregulation compared to T2P, “Down”: downregulation compared to T1P, 2a: in comparison to T2P as the comparison to T1P being not significant, N: no changes. 3: tumor (*n* = 290) in comparison to normal tissues (*n* = 30). 4: these genes are in Overlap21. 5: these genes are in Overlap21plus. Expression analysis in “CIMP” and “Tumor” using the TCGA data (UALCAN). *: *p* < 0.5, **: *p* < 0.01, ***: *p* < 0.001.

**Table 3 cancers-13-04483-t003:** Univariate and multivariate Cox analysis of Overlap66 and Overlap21 risk scores for pRCC OS.

Factors	Univariate Cox Analysis			Multivariate Cox Analysis		
	HR	95% CI	*p*-Value	HR	95% CI	*p*-Value
Overlap66	2.72	2.14–3.46	3.82 × 10^−16^ ***	3.03	2.29–4.01	1.15 × 10^−14^ ***
Overlap21	2.72	2.19–3.38	<2 × 10^−16^ ***	2.71	2.1–3.5	2.81 × 10^−14^ ***
Age	1.01	0.98–1.04	0.504	1.04 ^i^ 1.03 ^ii^	1.01–1.08 1.003–1.064	0.0119 * 0.0333 *
Sex	0.67	0.34–1.36	0.268	0.80 ^i^ 1.45 ^ii^	0.36–1.76 0.67–3.13	0.576 0.346
Tstage 1	5.13	2.73–9.62	3.53 × 10^−7^ ***	1.75 ^i^ 3.28 ^ii^	0.81–3.76 1.61–6.65	0.154 0.001 **

Analyses were performed using the TCGA PanCancer pRCC dataset. Age: at diagnosis. Sex: male vs. female. Tstage 1: T stage 3 + 4 vs. Tstage 0: T stage 1 + 2. i and ii: in analysis with Overlap66 (i) and Overlap21 (ii). *, **, and ***: *p* < 0.05, *p* < 0.01, and *p* < 0.001 respectively

## Data Availability

Data is present in this article and Appendix A.

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
