# Peer review of "Prognostic and Therapeutic Potential of the OIP5 Network in Papillary Renal Cell Carcinoma"

_cancers, 2021, doi:10.3390/cancers13174483_

Round 1

Reviewer 1 Report

In the current study, the authors analyzed the prognostic and therapeutic potential of the Opa interacting protein 5 (OIP5) in patient with papillary renal carcinoma (pRCC). Previous studies have shown that OIP5 can promote the initiation and progression of several cancer types. Overexpression of the OIP5 gene in ACHC pRCC cell line induced cell proliferation and invasion when compared to cells transfected with the vector only. Moreover, the analysis of gene expression in primary pRCC (n= 282) and ACHN OP5 xenograft tumors identified 66 shared genes, named overlap66. Interestingly, high-risk tumors can be stratified using either overlap66 or overlap21 identified risk genes.

Overall, the manuscript is well written and findings have clinical significance. However, the authors should address the following concerns:

  1. RNAseq gives a broad picture of the data, but the exact copy numbers of individual mRNAs need to be verified with a more sensitive method such qPCR for least some (10-20) genes.
  2. The changes in immune cells in pRCC shown in Fig. 8 is interesting. Again, the authors should validate the data using antibodies again known markers.
  3. Some of the data should be validated using new patients (not included in the experimental design).
  4. An antisense RNA that is transcribed in the antisense orientation of the gene encoding OIP5 was found to have an oncogenic function in multiple cancers such as cervical and lung cancers (Yang J et al. 2019, J Cell Biochem 120: 907-916; Zhang Z 2018, Biomed Pharmacother 101:14-23). What is the function of this antisense RNA in kidney cancer?

Author Response

We appreciate the reviewer’s overall positive tone and insightful remarks. Here are our detailed revisions.

“RNAseq gives a broad picture of the data, but the exact copy numbers of individual mRNAs need to be verified with a more sensitive method such qPCR for least some (10-20) genes.”

Authors' response – We agree with the reviewer for a need to confirm differentially expressed genes (DEGs) using real-time PCR. In our initial submission, PLK1 was confirmed; in this revision, we have made a major effort to confirm the upregulations of LYPD6 and PCSK5 (two component genes in Overalp66) in ACHN OIP5 tumors by real-time PCR. The data is presented and discussed (page 20, lines 526-530, marked with red). We trust Reviewer #1 sees the challenges facing to research caused by the current and lengthy pandemic. Confirmation of additional candidates will take substantial efforts at least time wise, as materials will take months to arrive in addition to the research restrictions enforced by Institute. Importantly, we don’t feel without confirmation of more DEGs will undermine the theme of this study, as 1) DEGs in Overlap66 were resulted from RNA-sequencing of 3 EV tumors and 3 OIP5 tumors as well as overlapping with OIP5-related DEGs in primary pRCC tumors, and 2) PLK1 possesses intriguing therapeutic values towards pRCCs with OIP5 upregulation. We hope the reviewer would agree with us.

“The changes in immune cells in pRCC shown in Fig. 8 is interesting. Again, the authors should validate the data using antibodies again known markers.

Some of the data should be validated using new patients (not included in the experimental design).”

Authors' response – We agree with the reviewer on both comments. Examination of either requires major efforts; it might be better to leave them in future research. We hope Reviewer #1 will see out points.

“An antisense RNA that is transcribed in the antisense orientation of the gene encoding OIP5 was found to have an oncogenic function in multiple cancers such as cervical and lung cancers (Yang J et al. 2019, J Cell Biochem 120: 907-916; Zhang Z 2018, Biomed Pharmacother 101:14-23). What is the function of this antisense RNA in kidney cancer?”

Authors' response – We thank Reviewer #1 for these insightful comments. It is an indeed intriguing issue regarding a relationship between OIP5 and OIP5-AS1 (OIP5 antisense RNA 1). OIP5-AS1 is a gene adjacent to the OIP5 bene. OIP5-AS1 encodes a long non-coding RNA (lnRNA) and thus affects oncogenesis via regulating miRNAs. Its regulatory relationship to OIP5 remains to be explored and its contributions to pRCC have not been reported. However, as the reviewer pointed out, OIP5 and OIP5-AS1 both can stimulate oncogenesis. In this regard, potential connections between both should be discussed. In this revision, a paragraph has been added to discuss this aspect (page 22, lines 597-612, marked with red) with the two articles cited by Reviewer #1 discussed.

Reviewer 2 Report

The study reveals important Information About OIP5 in pRCC. The manuscript includes a Variety of experimental studies to confirm the relevance of OIP5, includig TCGA data, immunohistochemical staining of Primary tissue, in vitro and in vivo studies. I recommend acceptance after few minor revisions:

  • include a Table showing the clinical Parameters of patients studied using a TMA
  • add multivariate analyses for the TCGA analysis

Author Response

We appreciate the reviewer for his/her encourage comments. Here are our detailed revisions.

 “include a Table showing the clinical Parameters of patients studied using a TMA”

 Authors' response – Thanks for the comment. A new table (Table 1) has been organized to present the clinical parameters (page 5, line 218, marked with red). Table 1 has bee discussed (Page 5, lines 200-202, marked with red). All “Table 1” terms in the last submission were changed to “Table 2” (marked with red).

“add multivariate analyses for the TCGA analysis”

 Authors' response – Multivariate analyses for Overlap66 and Overlap21 using the TCGA dataset have been included (Table 3; page 16, line 468, marked with red) and discussed (page 16, lines 444-445, marked with red).

Round 2

Reviewer 1 Report

Although no significant data were added, the manuscript can be acccepted for publication.